Salivary response of Geoffroy’s spider monkeys (Ateles geoffroyi) to consumption of plant secondary metabolites

Ramírez-Torres Carlos Eduardo ramireztcarlose@gmail.com neri3838@gmail.com 1
Gómez Fabiola Carolina Espinosa 2 3
Morales-Mávil Jorge E. 1
Mendoza-López María Remedios 4
Laska Matthias 5
Hernández-Salazar Laura Teresa herlatss@gmail.com 1
1 Instituto de Neuro-Etologia, Universidad Veracruzana , Xalapa , Veracruz , Mexico
2 Facultad de Medicina Veterinaria y Zootecnia, Universidad Popular Autonóma del Estado de Puebla , Puebla , Puebla , Mexico
3 Consejo de Ciencia y Tecnología del Estado de Puebla, Puebla, México , Puebla , Puebla , Mexico
4 Instituto de Química Aplicada, Universidad Veracruzana , Xalapa , Veracruz , Mexico
5 Department of Physics, Chemistry and Biology, IFM, Biology, Linkoping University , Linkoping , Linkoping , Sweden
Jimenez Manuel
Electronic publication date: 2025 Jun 6
Publication date: 2025
Volume: 13
Electronic Location ID: e19354
Received 2024 Dec 16; Accepted 2025 Apr 1
Copyright: ©2025 Ramírez-Torres et al.
Copyright year: 2025
Copyright holder: Ramírez-Torres et al.
License: This is an open access article distributed under the terms of the Creative Commons Attribution License, which permits unrestricted use, distribution, reproduction and adaptation in any medium and for any purpose provided that it is properly attributed. For attribution, the original author(s), title, publication source (PeerJ) and either DOI or URL of the article must be cited.
License URL: https://creativecommons.org/licenses/by/4.0/

Keywords: Frugivorous primates, Bitter compounds, Salivary proteins, Salivary pH, Proline-rich proteins

Funding: Secretaría de Ciencia, Humanidades, Tecnología e Innovación CERT, 931446 This work was supported by Secretaría de Ciencia, Humanidades, Tecnología e Innovación (CERT, 931446). The funders had no role in study design, data collection and analysis, decision to publish, or preparation of the manuscript.

==============================
Geoffroy’s spider monkeys (Ateles geoffroyi) can modulate the acidity-alkalinity (pH) and salivary expression of total proteins (TP) and proline-rich proteins (PRPs) depending on the concentration of tannins in their diet, helping to counteract negative post-ingestive effects. Besides tannins, plants produce a wide variety of secondary metabolites like flavonoids and alkaloids that elicit a bitter taste. Geoffroy’s spider monkeys feed on various plant species and consume different concentrations of secondary metabolites. However, it is unclear whether there is salivary modulation of pH, TP, and PRPs to secondary metabolites other than tannins, or whether this effect also occurs towards bitter substances not associated with secondary metabolites. Therefore, we assessed if there are changes in salivary pH, TP, and PRPs expression towards bitter substances or if spider monkeys display a specific response to secondary metabolites present in their diet and substances not associated with secondary metabolites. We determined the concentration of tannic acid, caffeine and rutin in fruits and leaves in different maturity stages reported as a part of the diet of Geoffroy’s spider monkeys. We presented six adults Geoffroy’s spider monkeys with different concentrations of tannic acid, caffeine, and rutin (0.1, 0.3, 0.6 and one mM) and denatonium benzoate (0.001, 0.003, 0.006 and 0.01 mM) dissolved in a 30 mM sucrose solution. We administered each concentration and collected saliva using swabs (SalivaBio). We used test paper strips to measure the pH and determined the TP concentration using the Bradford method at 595 nm. We also determined the percentage of PRPs using sodium dodecyl sulfate–polyacrylamide gel electrophoresis (SDS-PAGE). The results showed marked differences in tannic acid, caffeine and rutin concentration depending on the plant part and species. We found an increase in salivary pH in response to consumption of secondary metabolites, no variations in TP concentration, variations in the percentage of PRPs associated with tannic acid concentrations, and no significant changes when the animals consumed denatonium benzoate. Our results showed that spider monkeys specifically modulate acidity-alkalinity towards secondary metabolites and salivary PRPs expression towards tannic acid in their diet, and that they do not have a generalized salivary response to bitter compounds that are typically considered as toxic substances.

Introduction

Primates use plants as an essential food source because they contain nutrients such as carbohydrates, lipids, and proteins (Mattes, 2011; Lambert & Rothman, 2015). However, compounds found in certain plants or plant parts can be nutritionally limiting or even toxic, such as secondary metabolites, whose function is to protect plants from herbivory (Akula & Ravishankar, 2011; Barbehenn & Constabel, 2011; War et al., 2012; Villalba, Costes-Thiré & Ginane, 2017; Kariñho-Betancourt, 2018; Yang et al., 2018). The type of secondary metabolites varies depending on the plant species (Wink, 2003; Wink, 2008), and their concentration depends on the plant part and stage of maturity. For example, they are often found in higher concentration in young leaves and unripe fruits compared to ripe fruits, and during the ripening process of fruits, the concentration of organic acids systematically varies (Bashir & Abu-Goukh, 2003; Del Bubba et al., 2009; Sun et al., 2010; Huang, Li & Yang, 2012; Da Silva et al., 2014).

Secondary metabolites generally produce a bitter taste and an astringent sensation, and can generate negative post-ingestive effects such as nausea, abdominal pain and liver and kidney damage (Frutos et al., 2004; War et al., 2012; Stevenson, Nicolson & Wright, 2017). However, at low concentrations, some secondary metabolites can also act as antioxidants and immunostimulants (Crozier, Jaganath & Clifford, 2009).

The presence of secondary metabolites has been documented in the diet of all major primate taxa, including great apes (Rogers et al., 1990; Takemoto, 2003; Beaune et al., 2017), strepsirrhines (Carrai et al., 2003; Norscia, Carrai & Borgognini-Tarli, 2006; Norscia, Ramanamanjato & Ganzhorn, 2012), catarrhines (Chapman & Chapman, 2002; Fashing, Dierenfeld & Mowry, 2007; Ta et al., 2018) and platyrrhines (Felton et al., 2009; Espinosa-Gómez et al., 2018). In addition, most primate studies have focused on identifying the presence and concentration of tannins and alkaloids (Rogers et al., 1990; Hamilton & Galdikas, 1994; Wakibara et al., 2001; Chapman & Chapman, 2002; Norscia, Ramanamanjato & Ganzhorn, 2012; Ta et al., 2018; Thurau, Rahajanirina & Irwin, 2021), without considering other compounds such as flavonoids and terpenes. Considering the role that secondary metabolites play in the diet of primates (Glander, 1978; Rogers et al., 1990; Leighton, 1993; Kar-Gupta & Kumar, 1994; Fashing, Dierenfeld & Mowry, 2007; Windley et al., 2022), it is relevant to assess the relationship between specific secondary metabolites present in their diet and their physiological response (Windley et al., 2022).

Some primate species appear not to be physiologically affected by the secondary metabolites in the plants they consume (Oates, Waterman & Choo, 1980; Rothman, Dusinberre & Pell, 2009), although psychophysical studies show that they are able to perceive them, particularly tannic acid and caffeine (Laska et al., 2000; Laska, Rivas Bautista & Hernandez Salazar, 2009). This suggests that primates have physiological defense mechanisms that counteract the negative effect of these compounds, for example, modulation in salivary physicochemical characteristics, such as pH and proteins (Ramírez-Torres et al., 2022). Similarly, changes in the salivary concentration of proline-rich proteins (PRPs) and histatins allows them to capture and precipitate tannins and alkaloids, preventing them from interacting with proteins necessary for digestion (Oates, Waterman & Choo, 1980; Shimada, 2006; Fashing, Dierenfeld & Mowry, 2007; Morzel, Canon & Guyot, 2022; Windley et al., 2022).

Various studies indicate that non-human primates possess salivary PRPs, such as macaques (Macaca fascicularis and Macaca arctoides), baboons (Papio hamadryas) and howler monkeys (Alouatta palliata mexicana and Alouatta pigra) (Oppenheim, Kousvelari & Troxler, 1979; Schlesinger, Hay & Levine, 1989; Mau et al., 2011; Espinosa-Gómez et al., 2015; Espinosa-Gómez et al., 2018; Espinosa-Gómez et al., 2020). Similarly, histatins have been identified in the saliva of long-tailed macaques (Macaca fascicularis), gorillas (Gorilla gorilla gorilla), gibbons (Nomascus leucogenys), vervet monkeys (Cercopithecus aethiops) and langurs (Presbytis cristata) (Xu et al., 1990; Padovan et al., 2010). The specific dietary needs explain the physiological strategy based on salivary changes.

Geoffroy’s spider monkeys can modulate their salivary pH, total protein (TP) concentration, and PRPs percentage when consuming tannic acid (Ramírez-Torres et al., 2022). Geoffroy’s spider monkeys are primates that feed on more than 250 plant species, of which they mainly consume ripe fruits, although they may also feed on unripe fruits as well as on other plant parts such as leaves, flowers, and seeds (González-Zamora et al., 2009; Schaffner et al., 2012; Pablo-Rodríguez et al., 2015; Hartwell et al., 2021). This dietary diversity exposes them to different types and concentrations of secondary metabolites and to variations in the acidity of the nutrients that make up their diet. While secondary metabolites have not been identified in the plants consumed by Geoffroy’s spider monkeys so far, it is likely that they are present in their diet, as these monkeys can perceive the bitterness produced by such compounds (Laska, Rivas Bautista & Hernandez Salazar, 2009). However, it is unclear if this salivary response is specific to tannins or is a general response towards other secondary metabolites, such as flavonoids, alkaloids or synthetic bitter compounds. Therefore, we addressed the following questions: which secondary metabolites are present in the plants reported as consumed by Geoffroy’s spider monkeys? What are the salivary responses of spider monkeys to ingestion of alkaloids and flavonoids? Is there a general salivary response of spider monkeys to bitter stimuli?

Materials and Methods

Ethics statement

The experiments reported here comply with the Guidelines for the care and use of mammals in neuroscience and behavioral research (National Research Council, 2011), the American Society of Primatologists’ Principles for the Ethical Treatment of Primates, and Mexican laws (NOM-062-ZOO-1999 and NOM-051-ZOO-1995). The protocol was approved by the Secretaría de Medio Ambiente y Recursos Naturales (SEMARNAT; official permit numbers SGPA/DGVS/000041/22 and SPARN/DGVS/01767/22).

Collection of plant parts

We collected samples of each of the plant species listed below between February and October 2022. Each plant species was sampled from the middle strata of the trees. We estimated the maturity stage using the criteria of color, size and consistency to differentiate between unripe and ripe fruits, as well as between young and mature leaves on the same plant part (Di Fiore, Link & Dew, 2008; Felton et al., 2010; Felton et al., 2009). We collected 500 gr wet weight of unripe fruits (UF), ripe fruits (RF), young leaves (YL), and mature leaves (ML) from 64 trees of nine species reported as part of the diet of Geoffroy’s spider monkeys in the region of Los Tuxtlas, Veracruz, Mexico. The species collected were: (4) Ficus americana, (3) F. colubrinae, (3) F. rzedowskiana, (5) F. insipida, (8) F. yoponensis, (11) Spondias mombin, (7) S. radlkoferi, (12) Brosimum alicastrum and (11) Poulsenia armata. These species represent more than 40% of the spider monkey’s diet (González-Zamora et al., 2009).

Secondary metabolite analysis

Once the plant parts were collected, they were dehydrated until they reached a constant weight. We used a dehydrator (SANGKEE®) for the fruits at 35 °C and the leaves were dehydrated in a cardboard box (50 cm × 50 cm) under a 70-watt incandescent bulb. The dehydrated samples were ground to a fine powder. One pool was formed for each plant part (UF, RF, YL, and ML) per plant species. We took five g of each pool and added five ml of HPLC-grade methanol (1:1). Each mixture was vortexed for one minute and allowed to stand for 5 min. We took the supernatant using a five ml syringe and a 0.045 µm pore nylon filter placed at the bottom of the syringe to recover the supernatant after passing through the filter.

Once the sample was filtered, we determined the presence and concentration of tannic acid (tannin), caffeine (alkaloid), and rutin (flavonoid) using high-performance liquid chromatography (HPLC) Agilent Technologies HPLC System (model: 1,200 infinity). HPLC-grade methanol was used as a solvent at a temperature of 20 °C with a mobile phase of acetonitrile, methanol, and water (15:15:70) with a flow rate of 1 ml/min. The concentrations of the secondary metabolites were quantified using the HPLC equipment’s LC-UV-100 UV/VIS detectors. We performed the secondary metabolites analyses at the Institute of Applied Chemistry, Universidad Veracruzana.

Site and study subjects

We performed the behavioral tests at the Environmental Management Unit “Hilda Ávila de O’Farrill” of the Universidad Veracruzana, located near Catemaco in Los Tuxtlas, Veracruz, Mexico. 18°28′ and 18°26′N, 95°03′ and 95°01′W. We worked with three female and three adult male Geoffroy’s spider monkeys (n = 6). During the tests, the animals were housed in individual enclosures (4  ×  4  ×  4 m). When individuals completed the test, the sliding doors between the enclosures were opened, allowing them to interact with each other. The enclosures were enriched with trunks and vines to promote animal welfare. All individuals were exposed to natural conditions including temperature, relative humidity, and light-dark cycle. The monkeys were fed on cultivated fruits and vegetables once daily, and all tests were conducted two hours before their feeding time to ensure they were motivated to participate.

Study design

Each individual was presented with one bottle with a metal drinking spout, containing 100 ml of sucrose at a concentration of 30 mM (control solution) or an experimental solution containing the secondary metabolites tannic acid, rutin, or caffeine at concentrations of 0.1, 0.3, 0.6, and one mM, mixed with sucrose (30 mM). We used solutions of denatonium benzoate, a bitter-tasting synthetic compound, at concentrations of 0.001, 0.003, 0.006 and 0.01 mM mixed with sucrose (30 mM) (Fig. 1). All monkeys had access to one bottle for one minute (control or experimental). The sucrose concentration used is above the taste preference threshold of spider monkeys, which makes it attractive to induce their consumption, however, this concentration is low enough to avoid masking effects to other taste substances (Laska, Carrera Sanchez & Rodriguez Luna, 1996; Laska et al., 2000). The concentrations of the secondary metabolites and of the artificial bitter tastant were above the spider monkeys’ taste threshold (Laska et al., 2000; Laska, Rivas Bautista & Hernandez Salazar, 2009), which ensured that the individuals could reliably perceive them. The different solutions and saliva collection were conducted between 09:00 and 11:00 h. Sucrose (CAS# 57-50-1) was obtained from Merck, tannic acid (CAS# 1401-55-4) from Meyer, rutin (CAS# 207671-50-9), caffeine (CAS# 58-08-2) and denatonium benzoate (CAS# 3734-33-6) from Sigma-Aldrich.

Figure 1 Experimental design.

We administered each substance to all individuals for 49 days. In each period, the monkeys were initially offered the control solution containing 30 mM of sucrose and then, in ascending order, the four concentrations of tannic acid, rutin, caffeine (0.1, 0.3, 0.6 and 1 mM) or denatonium benzoate (0.001, 0.003, 0.006 and 0.01 mM) mixed with 30 mM of sucrose. Each concentration was offered for eight days, and on days six, seven and eight of each period salivary pH was recorded, and saliva samples were taken. After finishing the administration of a substance in its different concentrations, a three-day rest period was given, during which the control solution was offered so that the analysis would reflect the result of the concentration of each solution and not be an accumulated response. At the end of each period of administration of the substances, a two-week break was implemented, after which a new experimental period began.

Saliva collection

We collected saliva samples using a swab (SalivaBio Children’s Swab, Salimetrics 5001.08, SalivaBio, State College, PA, USA) with a length of 4.3 cm to allow the complete introduction of the swab into the monkey’s oral cavity. To encourage the monkeys to chew the swab, they were soaked with 0.5 ml of corn syrup for 60 s following the method reported by Ramírez-Torres et al. (2022).

Determination of salivary acidity/alkalinity (pH)

We determined salivary pH using colorimetric paper strips and compared the color of the test strips (OF®) with the standards (Ramírez-Torres et al., 2022).

Saliva sample processing

A pool was formed from the saliva obtained from each individual on days six to eight by exposing them to the different concentrations of the secondary metabolites and denatonium benzoate. We processed the samples following the method used by Espinosa-Gómez et al. (2018). We performed the saliva analysis at the Veterinary School of the Universidad Popular Autónoma del Estado de Puebla (UPAEP, University).

Determination of the total protein concentration (TP)

The Bradford method uses a spectrophotometer to measure the absorbance of total proteins at 595 nm (Bradford, 1976).

Determination of the presence of proline-rich proteins (PRPs)

We determined the presence of PRP by identifying protein bands in one-dimensional SDS-PAGE gel electrophoresis (Beeley et al., 1996), with modifications by Espinosa-Gómez et al. (2018). Once the proteins were fixed on the gels, they were stained using a Coomassie-R250 bath for four hours and washed out with 10% acetic acid baths. We used three µl of the molecular weight marker Flash Protein Ladder (Gel Company), which was loaded into the first lane.

Densitometric analysis of protein bands identified as proline-rich proteins (PRP) on 1D-sodium dodecyl sulfate–polyacrylamide gel electrophoresis (SDS-PAGE) gels

The electrophoresis gels were scanned at 1,200 dpi quality by HP Digital Sender Flow 8500 fn2 scanner to calculate the percentage of PRPs (% PRPs). Densitometry analysis of the gel images was performed using IMAGEJ software (Tiago & Wayne, 2012) following the method reported by Ramírez-Torres et al. (2022).

Statistical analysis

A factorial analysis was performed to determine if there were significant differences in each of the salivary physicochemical characteristics recorded for the different concentrations of the substances. Post hoc Tukey tests were performed using the R program version 4.1.1 to determine where these differences were found. We conducted all analyses using the statistical software R (R Core Team, 2023). We used the aov library to make the factor analysis of each physicochemical characteristic. Additionally, we employed the TukeyHSD library to make multiple comparisons of each physicochemical characteristic in relation to compound and concentration.

Results

Plants secondary metabolite analysis

The presence of tannic acid, caffeine, and rutin was identified in the plant species reported as part of the diet of Geoffroy’s spider monkeys, except caffeine in Ficus americana. Variations in the concentration of these secondary metabolites were found depending on the leaves and fruits and their maturity stage (Table 1). The highest concentration of tannic acid (252.43 µg/g) was recorded in the young leaves of F. insipida, ripe fruit of P. armata had the highest concentration of caffeine (5.26 µg/g), and mature leaves of F. americana presented the highest concentration of rutin (247.24 µg/g).

Table 1 Secondary metabolites in plants reported as consumed by spider monkeys (Ateles geoffroyi).

Concentration of tannic acid, caffeine and rutin in plant parts of the nine species reported as consumed by spider monkeys in Los Tuxtlas, Veracruz, Mexico.

Species	Plant part	Concentration (µg/g)	
		Tannic acid	Caffeine	Rutin	
Spondias mombin	Young leaves	–	–	–	
Mature leaves	–	–	–	
Unripe fruits	–	0.16	–	
Ripe fruits	3.85	1.8	0.38	
Spondias radlkoferi	Young leaves	–	–	22.5	
Mature leaves	2.28	–	1.1	
Unripe fruits	5.86	–	1.51	
Ripe fruits	16.72	0.65	5.15	
Ficus americana	Young leaves	–	–	0.14	
Mature leaves	1.1	–	247.24	
Unripe fruits	–	–	4	
Ripe fruits	–	–	–	
Ficus colubrinae	Young leaves	2.27	–	21.21	
Mature leaves	3.81	–	10.47	
Unripe fruits	0.58	0.05	0.23	
Ripe fruits	2.09	–	3.2	
Ficus insípida	Young leaves	252.43	–	1.94	
Mature leaves	8.81	–	0.84	
Unripe fruits	–	–	–	
Ripe fruits	5.19	5.15	0.17	
Ficus rzedowskiana	Young leaves	6.71	0.44	27.96	
Mature leaves	0.3	0.49	26.97	
Unripe fruits	3.19	0.74	1.92	
Ripe fruits	0.38	1.7	1.31	
Ficus yoponensis	Young leaves	–	–	18.14	
Mature leaves	–	–	19.22	
Unripe fruits	8.27	3.46	10.37	
Ripe fruits	14.47	0.41	0.95	
Brosimum alicastrum	Young leaves	–	–	–	
Mature leaves	16.01	–	–	
Unripe fruits	13.28	0.63	18.22	
Ripe fruits	12.08	1.82	19.29	
Poulsenia armata	Young leaves	0.8	0.41	0.38	
Mature leaves	3.54	0.79	–	
Unripe fruits	–	–	–	
Ripe fruits	4.15	5.26	1.72	

Salivary pH

Factor analysis indicated that there were statistically significant differences in the salivary pH of spider monkeys by compound (F = 21.72, p < 0.05), by concentration (F = 57.38, p < 0.05) and in the interaction between concentration and compounds (F = 18.05, p < 0.05). Tukey’s test identified the differences in salivary pH when the spider monkeys consumed the different concentrations of the solutions of secondary metabolites (Table 2). There was a significant increase in salivary pH with increasing concentrations of tannic acid and caffeine. We did not find significant differences in salivary pH with increasing the concentration of rutin and denatonium benzoate, respectively. The most alkaline salivary pH was 8.1 ± 0.25, recorded when the monkeys consumed the one mM concentration of tannic acid.

Table 2 Comparison of salivary pH of spider monkeys (Ateles geoffroyi).

Comparisons of salivary pH values of spider monkeys when offered different concentrations of secondary metabolites and denatonium benzoate. We put < when the pH of the X axis was significantly lower than that of the Y axis, while > when it was higher.

Substance		Tanic acid	Caffeine	
	Concentration
(mM)	0	0.1	0.3	0.6	1	0	0.1	0.3	0.6	1	
Tanic acid	0			<	<	<			<	<	<	
0.1				<	<				<		
0.3	>				<	>	>				
0.6	>	>			<	>	>				
1	>	>	>	>		>	>	>	>	>	
Caffeine	0			<	<	<			<	<	<	
0.1			<	<	<			<	<	<	
0.3	>				<	>	>				
0.6	>	>			<	>	>				
1	>				<	>	>				
Rutin	0				<	<				<	<	
0.1				<	<				<		
0.3				<	<				<	<	
0.6	>				<	>	>				
1	>				<	>	>				
Denatonium benzoate	0				<	<				<		
0.001				<	<				<		
0.003	>				<	>	>				
0.006	>				<	>	>				
0.01				<	<				<		

Total protein concentration

Factor analysis indicated significant differences in salivary TP of spider monkeys by compound (F = 6.35, p < 0.05). No significant differences were found by concentration or by the interaction of the compounds and their concentration. Tukey’s test identified significant differences between the TP concentration when monkeys consumed the 0.3 mM caffeine solution, and the control solution offered during the rutin administration period. There were significant differences in TP expression between the 0.3 mM concentration of caffeine and the 0.1 and 0.3 mM rutin concentrations.

Percentage of protein-rich proline

Factor analysis indicated statistically significant differences in the percentage of PRPs by compound (F = 71.24, p < 0.05), by concentration (F = 11.65, p < 0.05), and in the interaction between concentration and compounds (F = 6.43, p < 0.05). Tukey’s test identified the differences in the percentage of PRPs when the monkeys were offered different concentrations of solutions (Table 3). There was a significant increase in the percentage of PRPs with increasing concentrations of tannic acid. At the same time, there was no significant increase in the percentage of PRPs with increasing concentrations of caffeine, rutin or denatonium benzoate. The highest percentage of PRPs recorded was 27.63 ± 4.64, obtained when the monkeys consumed the one mM concentration of tannic acid.

Table 3 Comparison of % PRPs of spider monkeys (Ateles geoffroyi).

Comparison of PRPs percentage values of spider monkeys when offered different concentrations of secondary metabolites and denatonium benzoate. We put < when the PRPs percentage of the X axis was significantly lower than that of the Y axis, while > when it was higher.

Substance		Tanic acid	
	Concentration
(mM)	0	0.1	0.3	0.6	1	
Tanic acid	0		<	<	<	<	
0.1	>				 	
0.3	>				 	
0.6	>				 	
1	>	 	 	 	 	
Caffeine	0		<	<	<	<	
0.1		<	<	<	<	
0.3		<	<	<	<	
0.6			<	<	<	
1	 	<	<	<	<	
Rutin	0		<	<	<	<	
0.1		<	<	<	<	
0.3		<	<	<	<	
0.6		<	<	<	<	
1	 	<	<	<	<	
Denatonium benzoate	0		<	<	<	<	
0.001		<	<	<	<	
0.003		<	<	<	<	
0.006		<	<	<	<	
0.01	 	<	<	<	<	

We used the results of pH, TP and PRPs to show the pattern in the salivary response of the Geoffroy’s spider monkeys to the four bitter stimuli (Fig. 2).

Figure 2 Response of salivary physicochemical characteristics.

Variations in salivary pH, TP concentration and PRPs percentage of spider monkeys when consuming different. (A) Secondary metabolites (tannic acid, caffeine and rutin) and (B) denatonium benzoate.

Discussion

Secondary metabolites in plants consumed by Geoffroy’s spider monkeys

Our findings indicate that different types of secondary metabolites vary among plant species, with their concentrations fluctuating according to the specific part of the plant and its stage of maturity. Thus, our results confirm that spider monkeys are naturally exposed to secondary metabolites in their diet. We identified the presence of tannic acid, caffeine, and rutin in the fruits (both unripe and ripe) and leaves (young and mature) which are known to constitute a significant portion of the diet of Geoffroy’s spider monkeys in the Los Tuxtlas region (González-Zamora et al., 2009).

The rutin was found in the highest concentration on the plant parts, followed by tannic acid and caffeine. We found that the highest concentration of rutin was in mature leaves of F. americana, tannic acid in young leaves of F. insipida, and caffeine in ripe fruits of P. armata. Although there were variations in the concentrations of the three secondary metabolites between fruits and leaves, unlike what has been reported in the literature, ripe fruits did not always present lower secondary metabolite concentration (Sun et al., 2010; Huang, Li & Yang, 2012; Da Silva et al., 2014).

To the best of our knowledge, this is one of the first studies that delves into the identification and quantification of tannic acid, caffeine and rutin since most studies so far only focus on determining the presence and concentration of groups of substances such as tannins and alkaloids (Rogers et al., 1990; Hamilton & Galdikas, 1994; Wakibara et al., 2001; Chapman & Chapman, 2002; Norscia, Ramanamanjato & Ganzhorn, 2012; Ta et al., 2018; Thurau, Rahajanirina & Irwin, 2021; Li et al., 2022).

Psychophysical studies have used tannic acid and caffeine to evaluate the behavioral and physiological responses of western gorillas (Gorilla gorilla gorilla), pig-tailed macaques (Macaca nemestrina), squirrel monkeys (Saimiri sciureus) and Geoffroy’s spider monkeys (A. geoffroyi) to bitter or astringent substances (Critchley & Rolls, 1996; Remis & Kerr, 2002; Laska, Rivas Bautista & Hernandez Salazar, 2009; Ramírez-Torres et al., 2022). To our knowledge, this is also the first report of rutin in plants consumed by non-human primates. Rutin, a flavonoid that has been studied in functional foods, produces bitter flavor and astringent sensation (Kim, Kim & Hwang, 2023) and, like other flavonoids, has been shown to have effects as a glycemic level controller, anti-inflammatory, antioxidant and neuroprotective agent (Crozier, Jaganath & Clifford, 2009; Hosseinzadeh & Nassiri-Asl, 2014; Ghorbani, 2017; El-Beltagi et al., 2019). The consumption of rutin by spider monkeys could have a potentially beneficial effect, as has been described in female primates that consume them to solve stressful situations such as the postpartum period (Colombage et al., 2024).

Salivary response to tannic acid, caffeine and rutin

In primates, saliva acts as a modulator to reduce the harmful effects of secondary metabolites, altering its physicochemical characteristics such as pH and proteins (Fashing, Dierenfeld & Mowry, 2007; Espinosa-Gómez et al., 2020; Morzel, Canon & Guyot, 2022; Ramírez-Torres et al., 2022). The saliva of Geoffroy’s spider monkeys became alkaline when we offered higher concentrations of tannic acid and caffeine. We recorded the highest pH value in their saliva after the animals consumed the highest concentration of tannic acid.

Changes in salivary pH in Geoffroy’s spider monkeys when consuming different secondary metabolites are thought to act as a defense mechanism to avoid acid-associated damage from food (Lavy et al., 2012; Ramírez-Torres et al., 2022). The differential response in the salivary pH levels to the tested compounds could be due to the fact that they present different degrees of acidity; tannic acid (pH 3.5), caffeine (pH 4.4 to 5.5), rutin (pH 6.5 to 7), and denatonium benzoate (pH 6.8 to 7.8), whereby the body releases only the necessary amount of salivary alkalinity-promoting buffers such as bicarbonate ions, thus avoiding damage to the teeth and oral cavity associated with the organic acids in the food (Foley, McLean & Cork, 1995; Pedersen & Belstrøm, 2019). This has been reported in rhesus macaques (Macaca mulatta), squirrel monkeys (Saimiri sciureus) and Geoffroy’s spider monkeys (Ateles geoffroyi) (Pritchard, Bowen & Reilly, 1995; Laska, 1996; Laska, 1999; Laska et al., 2000). It has been proposed that this ability enables them to consume acids and essential amino acids in fruits and thus to assess their nutritional value (Larsson et al., 2014). In addition, these changes could contribute to the ability of Geoffroy’s spider monkeys to consume unripe fruits when ripe fruits are unavailable (Pablo-Rodríguez et al., 2015; Batista-Silva et al., 2018).

Conversely, we observed an increase in TP with higher concentrations of tannic acid consumed by Geoffroy’s spider monkeys; however, this increase was not pronounced. Our data revealed significant differences in TP concentration between caffeine and rutin. We did not find a significant increase in TP in Geoffroy’s spider monkeys with denatonium benzoate. This range was consistent across all substances tested. It is well-established that diet and taste sensations can influence salivary protein expression (Dawes & Shaw, 1965; Neyraud et al., 2006; Quintana et al., 2009; Morzel et al., 2012; Torregrossa et al., 2014).

Furthermore, in some primates, it has been shown that in addition to secondary metabolites, fiber also influences TP expression and concentration (Neyraud et al., 2006; Quintana et al., 2009; Canon et al., 2010; Ramírez-Torres et al., 2022). Although we did not estimate the fiber content in the plant parts, our study demonstrates that the bitter taste sensation elicited by the secondary metabolites is not sufficient to cause significant changes in the concentration of salivary TP in Geoffroy’s spider monkeys. Some studies showed higher levels of TP concentration in folivorous species like Alouatta palliata or granivorous primates such as Macaca arctoides, which is related to their dietary habits and the exposition to higher amounts of fiber (Milton, 1978; Dias & Rangel-Negrín, 2015; Masi et al., 2015; Thamadilok et al., 2019). Since spider monkeys are mainly frugivorous (Klein & Klein, 1977; Felton et al., 2010), this might be a salivary response based on their kind on specific dietary needs.

We found differences in the percentage of PRPs based on compound, concentration, and their interaction. The percentage of PRPs in the saliva increased when the monkeys consumed tannic acid from its lowest concentration, with the salivary expression of PRPs increasing at higher concentrations of this substance. However, no significant changes were recorded in the PRPs for caffeine and rutin. Similarly, no significant differences were recorded in the PRPs percentage with the different concentrations of denatonium benzoate.

PRPs are present in animals that include some plant parts in their diet and are absent in carnivores (McArthur, Sanson & Beal, 1995; Espinosa-Gómez et al., 2013; Espinosa-Gómez et al., 2015; Espinosa-Gómez et al., 2018; Espinosa-Gómez et al., 2020; Ramírez-Torres et al., 2022). This is in line with our results that agree with the literature that PRPs act as a defense mechanism against the consumption of tannins and that it is not a general protein expression (Shimada, 2006; Mau et al., 2011; Espinosa-Gómez et al., 2015; Espinosa-Gómez et al., 2018; Espinosa-Gómez et al., 2020; Morzel, Canon & Guyot, 2022; Windley et al., 2022). However, the production of this type of protein does not increase with the consumption of other secondary metabolites, such as alkaloids and flavonoids. The substance-specific salivary response of the monkeys may be linked to the diverse bitter-taste receptors which are sensitive to various molecules that elicit this taste quality (Matsunami, Montmayeur & Buck, 2000; DuBois, De Simone & Lyall, 2008; Kuhn et al., 2010). The differences in the chemical structures of tannins, alkaloids, and flavonoids lead to their perception by specific receptors, possibly resulting in varied responses in protein production. Additionally, tannins elicit astringency, as it has been shown to stimulate the production of PRPs (Kauffman & Keller, 1979; Wróblewski et al., 2001). This may explain why an increase in PRP levels was observed only when the monkeys consumed tannic acid, but not with caffeine or rutin. Accordingly, we propose that the expression of salivary PRPs may be a specific response to tannins. This does not rule out the existence of other defense responses against other plant secondary metabolites that prevent the damage associated with them, for example, histatin-rich proteins which are effective in capturing secondary metabolites (Naurato et al., 1999; Shimada, 2006) and the intestinal microbiota which produces a wide variety of enzymes with the ability to form a protective biofilm and detoxify potentially toxic compounds, such as secondary metabolites (McKey et al., 1981; Zhang et al., 2020; Xia et al., 2023). Furthermore, it is essential to highlight that no significant changes were found in the percentage of PRPs when presenting the Geoffroy’s spider monkeys with denatonium benzoate which is a synthetic bitter-tasting substance. Accordingly, we conclude that the expression of this type of protein does not occur in response to the perception of bitter taste as such (Frutos et al., 2004; War et al., 2012; Stevenson, Nicolson & Wright, 2017). Thus, Geoffroy’s spider monkeys, despite perceiving bitter taste as something negative, possess physiological mechanisms that allow them to react differentially and specifically to the wide variety of compounds that cause an avoidance response (Fig. 2). This physiological discrimination based on secondary metabolite type is essential for these primates, enabling them to be selective in obtaining readily metabolizable energy, but with the ability to include a wide variety of food species and plant parts, and to respond physiologically to compounds that do not represent the same adverse effects on their health, and whose consumption may even provide benefits (Schaffner et al., 2012; War et al., 2012; Pablo-Rodríguez et al., 2015; Kariñho-Betancourt, 2018; Yang et al., 2018; Hartwell et al., 2021).

Conclusion

This study demonstrates the presence of different secondary metabolites in plants consumed by spider monkeys and provides the first evidence of the flavonoid rutin in the diet of primates. The results support the hypothesis that spider monkeys have specific salivary mechanisms for certain secondary metabolites, such as tannic acid, while for other compounds, such as caffeine or rutin, salivary protein expression is not affected. Data obtained with the synthetic compound denatonium benzoate showed that the perception of bitter taste as such is not a general trigger for protein expression or changes in salivary acidity.

Supplemental Information

Supplemental Information 1 ARRIVE guidelines

Gildardo Castañeda and Alejandro Coyohua for their help during field work and Dr. Eduardo Reynoso and Dr. Serio Silva for the comments. Many thanks to all the monkeys that participated in our experiments.

Additional Information and Declarations

Competing Interests

Author Contributions

Animal Ethics

Data Availability

The authors declare there are no competing interests.

Carlos Eduardo Ramírez-Torres conceived and designed the experiments, performed the experiments, analyzed the data, prepared figures and/or tables, authored or reviewed drafts of the article, and approved the final draft.

Fabiola Carolina Espinosa Gómez conceived and designed the experiments, performed the experiments, analyzed the data, authored or reviewed drafts of the article, and approved the final draft.

Jorge E. Morales-Mávil conceived and designed the experiments, analyzed the data, authored or reviewed drafts of the article, and approved the final draft.

María Remedios Mendoza-López conceived and designed the experiments, performed the experiments, analyzed the data, authored or reviewed drafts of the article, and approved the final draft.

Matthias Laska conceived and designed the experiments, analyzed the data, prepared figures and/or tables, authored or reviewed drafts of the article, and approved the final draft.

Laura Teresa Hernández-Salazar conceived and designed the experiments, performed the experiments, analyzed the data, prepared figures and/or tables, authored or reviewed drafts of the article, and approved the final draft.

The following information was supplied relating to ethical approvals (i.e., approving body and any reference numbers):

The experiments reported here comply with the Guidelines for the care and use of mammals in neuroscience and behavioral research (National Research Council, 2011), the American Society of Primatologists’ Principles for the Ethical Treatment of Primates, and Mexican laws (NOM-062-ZOO-1999 and NOM-051-ZOO-1995). The protocol was approved by the Secretaría de Medio Ambiente y Recursos Naturales (SEMARNAT; official permit numbers SGPA/DGVS/000041/22 and SPARN/DGVS/01767/22).

The following information was supplied regarding data availability:

The data and code are available at figshare: Ramírez Torres, Carlos Eduardo (2024). Database and code Salivary response of spider monkeys. figshare. Dataset. https://doi.org/10.6084/m9.figshare.27961557.v1.

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
