# Peer review of "Salivary response of Geoffroy’s spider monkeys (Ateles geoffroyi) to consumption of plant secondary metabolites"

_PeerJ, doi:10.7717/peerj.19354_

## Round 0.1 · original submission · Major Revisions

Dear Co-Authors,

We have received a “reject” evaluation for the manuscript. However, I believe that the reviewer’s comments, while valid and accurate, are not insurmountable. Therefore, I have decided to reconsider it as “major revisions,” acting in my capacity as editor and as an additional reviewer, with the aim of improving the manuscript and leveraging my expertise in saliva diagnostics. The rejection appears to stem from structural and methodological issues that may affect the validity of the results and the replicability of the study. I kindly ask you to address the reviewers’ comments thoroughly and also revise the following critical points:

1. Review the manuscript’s structure: The introduction and discussion should be more coherently integrated. The discussion should be divided into two subsections to facilitate clarity and eliminate redundancies and repetitions.
2. Expand the description of the sampling and collection procedures: Could you provide more detail and clarity in these sections?
3. Clearly and comprehensively describe the connection between secondary metabolites and salivary responses: This linkage needs to be developed more explicitly.

Resolving these points and adequately responding to the reviewers’ comments will be crucial in determining whether your manuscript can be considered for publication in PeerJ.

Kind regards,
Dr. Manuel Jiménez

·

Basic reporting

This is a study on the plant secondary compounds and spider monkeys' physiological responses to its ingestion.

Concerning the first part of the study, concentration of the three different plant secondary compounds, what is mentioned in the discussion is very brief, and I am afraid that it does not match the lengthy discussion in the discussion section. For example, if the changes in maturity is important, it can be tested statistically, rather than just showing the raw data. I suggest the authors to present a number of testable predictions, based on what has been known about the protection strategy of plants, and examine those predictions based on their own data.

There is no direct link between the two parts of the study (secondary compounds concentrations in plants and spider monkeys' physiological responses) both in the results and discussion. I understand the first part can offer a background of the second part, but I would like to encourage the authors to explain more in the introduction why these two different kinds of data are necessary to explore the main question of this study. Just listing individual questions (LL104-108) is not enough. In my opinion, what might be possible is that the first part may offer the likely threshold value of the compounds above which physiological response occurs, because both parts show the data of the concentration of the compounds. I know that the units differ between the two parts (µg/g vs mM), but it is worth trying.

Experimental design

I think the method is solid and the discussion is reasonable and important.

Validity of the findings

I think the method is solid and the discussion is reasonable and important.

Additional comments

There is no subsection in the discussion, and it is very difficult to understand the overall structure of the discussion. Please divide it into some subsections and show the structure explicitly to readers.

LL89-90: crab macaque>crab-eating macaque, or recently more commonly called as long-tailed macaque

L35: Do not use an abbreviation (PRP) without showing the definition. It must appear on LL31-32.

L100: Same as above for TP. In addition, I am not sure why the authors consider TP is important. They introduce previous studies that salivary pH and RPR may work as a countermeasure for secondary compounds, but not for TP.

LL201-202: Indicate the package/function to perform the test.

L203: Cite Figure 2 somewhere in the results section.

LL253:256: It is not necessary clear from the context 'what has been reported in the literature' is. Consider revision.

L270: pigtailed>pig-tailed

LL268-272: It is awkward the two consecutive sentences start with 'However'. Consider revision.

·

Basic reporting

A series of minor observations are presented, all indicated in the PDF.
1) Review the common name of Ateles geoffroyi. The authors use one corresponding to a subspecies from Nicaragua or indicating the taxonomic source. This change does not modify the results.
2) The words "monkey" and "spider monkey" are used interchangeably; I suggest that the name in English be standardized throughout the text.
3) Use the complete name before to indicate its abbreviation.
4) Check that all references are in the text because some only appear in the References section.

Experimental design

No comment

Validity of the findings

This paper presents novel and rigorous research on the topic presented. All results are adequately presented, supported by tables, figures, and statistical analysis.

·

Basic reporting

Aim of the study was clear.
Literature references was adequate and updated.
Tables were well structured.
Well explained hypothesis

Experimental design

Yes the paper is within the scope of the journal
The research question was well defined, relevant and meaningful.
The research work filled the gaps and lacunae.
technical and ethical standard was maintained.
Descriptive methodology has been explained

Validity of the findings

Impact and novelty assessed.
Results of the study were impactful
All datd was robust and statistically sound.
Well structured manuscript and well stated conclusion was there

Additional comments

Following are the queries w.r.t manuscript:
1. What could be the possible explanation of physiological mechanism by which saliva of spider monkeys converts the toxic effect of the secondary metabolites into less toxic form?
2. What is the significance of this adaptation in the studied animals and why they depend on such toxic plant metabolites as they may feed on non toxic plants? Is that the feeding behaviour that develops this adaptation or the scarcity of the food in the environment which led these monkeys to depend on these plants secondary metabolites?
3. Is there any evolutionary trend for the development of this adaptation in these monkeys?
4. I appreciate the biochemical analysis for significant results.
5. There are certain grammatical and language errors which were highlighted in the attached pdf

Reviewer 4 ·

Basic reporting

logic of language is poor and need to be improved

Experimental design

the authors did not provided detailed information on the plant sample collection, which can influence their results.

Validity of the findings

the authors seems to tell us two different study results which were not related each other

Additional comments

This paper provided some basic information on the salivary response to black-handed spider monkeys to consumption of plant secondary metabolites. However, the results are descriptive. there are several problems need to be resolved. Firstly, the authors seems to tell us two different study results which were not related each other. Secondly, the authors did not provided detailed information on the plant sample collection, which can influence their results. Thirdly, logic of language is poor and need to be improved. It is difficult for understanding the content of Table 2. Detailed comments see below:
L1-2, which spider monkey species? In this paper, it is black-handed spider monkeys.
L23-44, the Abstract need to be rewritten. The author spent too much words on the background information and methods. But they did not provide the results on the secondary metabolite of the food plants?
L47, the logic of Introduction is very confused, and need to be written. For example, the paragraph (L85-92) is very abrupt. There was no correlation to the front and latter paragraphs. Moreover, the authors should give more information on the ecology of black-handed spider monkeys. Why did the authors select this species as study subject?
L104-108, The questions are not clear. For example, the first question is about secondary metabolites in food plants, including all secondary metabolites? The second question should be about the change in metabolites concentrations of plant parts with their growth.
117-123, the authors should give more detailed describe on the method of plant sample collection.
L154-157, How to determine the concentrations gradient? Is there preliminary experiment? How many times of experiment for each individual?
L133-134, why did the authors only test tannic acid, caffeine, and rutin?
L205-211, The authors should conduct the statistic tests on these variations.
L218-219, from Figure 2? But the salivary PH decreased between the caffeine concentrations of 0.6 and 1?
L227-231, from which Figures or Tables?
L237, is there statistic test for the significant increase?
L243, the Discussion is too long, and the authors spent much words on describing the results repeatedly. They should focus on the explanation on the significant differences.

---

## Round 0.2 · accepted · Accept

Dear Authors:

Congratulations, your manuscript: "Salivary response of Geoffroy´s spider monkeys (Ateles geoffroyi) to consumption of plant secondary metabolites" has been Accepted for publication.

Regards,

Dr. Manuel Jiménez

·

Basic reporting

I have confirmed the revisions and the authors' responses to my comments on the previous version. Even though not all of my comments are incorporated, I understand why the authors did not do so. I think the authors did the best they can.

Experimental design

No specific comments any more.

Validity of the findings

No specific comments any more.

Additional comments

No specific comments any more.

·

Basic reporting

No Comment

Experimental design

No Comment

Validity of the findings

No comment

Additional comments

The authors has successfully answered my queries